# Retrieving Volcanic Ash Top Height through Combined Polar Orbit Active and Geostationary Passive Remote Sensing Data

**Weiren Zhu [1], Lin Zhu [2],\*, Jun Li [3] and Hongfu Sun [1]**

[1] College of Geoscience and Surveying Engineering, China University of Mining and Technology, Beijing 100083, China; zqt1700203072g@student.cumtb.edu.cn (W.Z.); Sunhongfu@cumtb.edu.cn (H.S.)

[2] National Satellite Meteorological Center, China Meteorological Administration, Beijing 100081, China

[3] Cooperative Institute for Meteorological Satellite Study, University of Wisconsin–Madison, Madison, WI 53706, USA; jun.li@ssec.wisc.edu

\* Correspondence: zhulin@cma.gov.cn

**Abstract:** Taking advantage of both the polar orbit active remote sensing data (from the Cloud-Aerosol Lidar with Orthogonal Polarization—CALIOP) and vertical information and the geostationary passive remote sensing measurements (from the Spinning Enhanced Visible and Infrared Imager) with large coverage, a methodology is developed for retrieving the volcanic ash cloud top height (VTH) from combined CALIOP and Spinning Enhanced Visible and Infrared Imager (SEVIRI) data. This methodology is a deep-learning-based algorithm through hybrid use of Stacked Denoising AutoEncoder (SDA), the Genetic Algorithm (GA), and the Least Squares Support Vector Regression (LSSVR). A series of eruptions over Iceland's Eyjafjallajökull volcano from April to May 2010 and the Puyehue-Cordón Caulle volcanic complex eruptions in Chilean Andes in June 2011 were selected as typical cases for independent validation of the VTH retrievals under various meteorological backgrounds. It is demonstrated that using the hybrid deep learning algorithm, the nonlinear relationship between satellite-based infrared (IR) radiance measurements and the VTH can be well established. The hybrid deep learning algorithm not only performs well under a relatively simple meteorological background but also is robust under more complex meteorological conditions. Adding atmospheric temperature vertical profile as additional information further improves the accuracy of VTH retrievals. The methodology and approaches can be applied to the measurements from the advanced imagers onboard the new generation of international geostationary (GEO) weather satellites for retrieving the VTH science product.

**Keywords:** volcanic ash cloud top height; stacked noise reduction encoder; CALIOP radar data; geostationary satellite imager; retrieval algorithm; deep learning

## 1. Introduction

Volcanic eruptions are natural disasters that can strongly affect climate and aviation safety [1,2]. Volcanic ash cloud top height (VTH) is a key parameter in the transport models [3]. During the past two decades, both space-based and ground-based remote sensing techniques have been widely used to quantitatively estimate the height and concentration of volcanic ash clouds [4–11].

Ground-based remote sensing instruments, such as microwave weather radar, can gather three-dimensional information of atmosphere with a repeat cycle of few minutes and in all weather conditions [12]. Many techniques have been developed for retrieving the VTH from these ground-based remote sensing measurements [12,13], and these approaches typically offer very accurate estimates of VTH [14,15]. However, these ground-based instruments can only offer observations with spatial

coverage up to several hundreds of kilometers, which limits their further applications in large scale monitoring.

Compared with the ground-based instruments, space-based satellites (including both passive and active sensors onboard the satellites) offer observations with better spatial coverage. Generally, there are four methodologies for estimating the VTH using space-based remote sensing measurements [7,14], which includes the parallax method [8,14,16], the wind correlation method [17], the lidar method [18], and the infrared (IR) radiation method [4–6]. Active lidar method and parallax method have higher accuracy (with an accuracy better than 200 m). However, the spatial coverage and repeat cycle are limited for large-scale applications. Passive remote sensing from satellites, especially from the geostationary (GEO) satellites [5], on the other hand, provides multispectral observations for volcanic ash with large coverage and high spatial–temporal resolution, but with limited retrieval accuracy.

Since the 1990s, estimates of VTH have been achieved from satellite-based IR observations. By comparing laboratory measurements of the spectral absorptions of ice clouds, water clouds, and volcanic minerals, a "reverse absorption" signal was found for volcanic ash [10]. The signal is from the difference between the absorptions in two IR spectral channels, centered at 11 and 12 μm, and can be used to distinguish the volcanic ash from land surfaces and other meteorological clouds. Until now, the Split Window Temperature Difference (SWTD) algorithm, which is based on the reverse absorption signal, has been widely used for retrieving VTH from satellite-based IR observations [6,19]. However, the reverse absorption signal was found to be effective for volcanic ash retrieval only in laboratory measurements. When applied to real satellite remote sensing observations, many factors can influence the signal. For example, the temperature of the volcanic ash, ambient atmospheric conditions, atmospheric moisture content, atmospheric temperature, and surface emissivity with complicated spectral, temporal and spatial variations all influence the signal and may lead to failure of the SWTD method [4,7,20–22].

In addition to the two "split window" channels, there are also other channels that are found sensitive to the mineral composition and $SO_2$ in the volcanic cloud. For example, volcanic $SO_2$ absorbs strongly in both 7.3 um and 8.6 um spectral regions [23,24]. In addition, a 13.3 μm channel from the new generation of international GEO weather satellites [25,26] adds considerable sensitivity to clouds [5,27], which is closely related to VTH. Therefore, the traditional SWTD technique has been improved through the addition of observations from more IR channels for volcanic $SO_2$ and ash property retrieval [4,23,27,28]. Based on the combined use of forward radiative transfer model (RTM) and inverse theory, many algorithms have been developed for retrieving volcanic ash and $SO_2$ parameters (such as height and mass loading), those include the look-up table method [4,6,23,28], a one-dimensional variational (1DVAR) method [5,29–31], linear statistical regression methods [7], and neural network method [15,32].

Although those algorithms have been used for retrieving volcanic ash and $SO_2$ parameters in the past two decades, uncertainties remain. The retrieval uncertainties from those algorithms are attributed to many factors including observation errors (detector noise, geolocation, calibration and channel-to-channel coregistration, data processing, etc.), the uncertainty of ancillary data and the forward RTM, the assumption of cloud vertical structure, and the complicated nonlinear relationship between volcanic ash parameters and satellite-based IR observations, etc. [7,14,33–37]. It is estimated that about 40% of total mass retrieval error is attributed to the uncertainties of input parameters including atmospheric vertical profiles, plume geometry, surface temperature, IR emissivity, and ash type [28]. In addition, forward RTM calculations are also associated with significant retrieval uncertainties, especially when performed over land surfaces and under cloudy skies [5,38]. When the optical thickness of a volcanic ash cloud is low (such as for a semi-transparent cloud), more uncertainties will contribute to the calculated top-of-atmosphere (TOA) radiances. Furthermore, current RTMs used in operations have been limited to clear skies or single-layer cloud situations [39]. In case of vertically distributed multilayer clouds that are blocked by upper meteorological layers, forward

RTMs cannot accurately simulate the IR radiative transfer process, potentially resulting in large VTH retrieval uncertainties.

In summary, no single remote sensing system can give a comprehensive description of eruptive activity [11]. Recently, a multisensor approach for volcanic ash cloud retrieval was developed by integrating satellite-based passive satellite data and ground-based active weather radar [11], which provides a way for using multisensor measurements. In our work, data from both passive remote sensing from GEO and active remote sensing from polar orbit (LEO) satellites are combined using a hybrid deep learning method for deriving the VTH. The uniqueness of this study consists in the following: (1) combining active remote sensing data from LEO with vertical information and passive remote sensing measurements from GEO with large coverage and high spatial–temporal resolution, and (2) using machine learning techniques to handle the nonlinearity between satellite-based IR radiances and VTH and avoid the RTM uncertainties in ash cloudy situations, especially in multilayer cloudy skies. Three machine learning methods, the Stacked Denoising AutoEncoder (SDA), the Genetic Algorithm (GA), and the Least Squares Support Vector Regression (LSSVR), are used to remove redundant information among IR channels and solve the multiconstraint nonlinear problem. The descriptions of those machine learning algorithms are given in the Appendix A. A series of eruptions over Iceland's Eyjafjallajökull volcano from April to May 2010, and the eruptions of the Puyehue-Cordón Caulle volcanic complex (PCCVC) in Chilean Andes in June 2011, were chosen as typical cases for independent validation of VTH retrievals.

Section 2 introduces the data used in this study. Section 3 describes the methodology based on the hybrid use of the SDA, the GA, and the LSSVR, for retrieving VTH. Independent validation on VTH retrievals from the Eyjafjallajökull and PCCVC cases is given in Section 4. The VTH retrieval sensitivity to the atmosphere temperature profile input is analyzed in Section 5, and the VTH retrieval sensitivity to feature selection is investigated in Section 6. Uncertainty analysis is given in Section 7. Section 8 contains the concluding remarks. In addition, the detailed descriptions of the machine learning algorithms used in this study are given in Appendix A.

## 2. Data

### 2.1. SEVIRI L1 Data

Brightness temperature (BT) from Spinning Enhanced Visible and Infrared Imager (SEVIRI) IR channels, the channel 4 (3.9 μm), 5 (6.25 μm), 7 (8.7 μm), 9 (10.8 μm), 10 (12.0 μm), and 11 (13.4 μm), were used to build a volcanic ash matchup dataset for training and testing the VTH retrieval model. SEVIRI is a 12-channel imager onboard the Meteosat Second Generation (MSG) with a repeat cycle of 15 min [40]. SEVIRI data corresponding to the eruptions of interest were downloaded from the European Organization for the Exploitation of Meteorological Satellites' (EUMETSAT) Earth observation portal (https://eoportal.eumetsat.int/userMgmt).

The main physical basis behind the discrimination of volcanic ash in thermal IR observations is that silicate ash has the opposite spectral signature to water/ice clouds in the thermal IR split window (which is defined by spectral channels centered on wavelengths near 11 and 12 μm). Based on the difference between radiances recorded in the split window channels, the height and concentration of the VTH can be estimated by simulating the radiative transmission through the cloud. Any $SO_2$ gas present in the volcanic ash cloud absorbs strongly around 8.5 μm, where a SEVIRI channel is centered [41,42]. Furthermore, SEVIRI channel 11 (centered on 13.4 μm) is sensitive to cloud radiative temperature [27]. To maximize the utilization of information for deriving VTH, data from six thermal IR channels were included in training the retrieval model and verifying VTH results.

### 2.2. CALIOP Data

VTH derived from CALIOP L2 cloud top height products with 5 km horizontal resolution was used as a "true" data for both training and independent validation. CALIOP is an active Cloud-Aerosol

Lidar, developed jointly by the National Aeronautics and Space Administration (NASA) and the French Space Agency (CNES). It is the main instrument onboard the Cloud-Aerosol LIDAR and Infrared Pathfinder Satellite Observation (CALIPSO) platform [18,43,44]. The CALIPSO platform was launched, together with the CloudSat satellite, on 28 April 2006. CALIPSO is a key satellite within NASA's A-train constellation which also includes Aqua, CloudSat, Parasol, and Aura [45]. CALIOP can provide vertical profiles of clouds and aerosols by measuring backscatter signals at 532 nm (both parallel and perpendicular) and at 1064 nm [43]. The vertical sampling resolution of CALIOP is 30 m under 8.2 km and 60 m from 8.2 to 20.2 km. The horizontal resolution is 333 m in the lower troposphere. Three types of profiles are provided in CALIOP level 1B product: total backscatter (parallel plus perpendicular) at 535 and 1064 nm and the 532 nm perpendicular backscatter. In addition, there are three basic types of level 2 data products derived from CALIOP at various spatial resolutions, including layer products, profile products, and the vertical feature mask (VFM) [43]. In this study, the CALIOP level 1B data (532 nm total attenuated backscatter product with 333 m horizontal resolution in the lower troposphere) were also used to analyze the vertical structure of volcanic cloud and distinguish meteorological cloud and volcanic ash. CALIOP Level 1B and Level 2 data are available from the NASA Langley Research Center Atmospheric Science Data Center (ASDC) website (http://reverb.echo.nasa.gov/reverb).

### 2.3. Atmospheric Profile Data

To account for the influence of atmospheric temperature on the observed radiances, the temperature vertical profiles from the European Centre for Medium-Range Weather Forecasts' (ECMWF) reanalysis data, ERA-Interim, were colocated with the training samples in the training data set. ERA-Interim data are derived from a much improving atmospheric model and assimilation scheme, relative to those used for the ERA-40 reanalysis. ERA-Interim data are provided on a grid with a horizontal resolution of 80 km and 37 vertical levels from the surface to 0.1 hPa. The products include global three-hourly surface parameters, including ocean-wave and land-surface conditions, and six-hourly upper-air parameters [46]. For this study, temporally and spatially interpolated temperature profiles should be used. The data are provided in 37 vertical layers, distributed according to changes in air pressure. The ERA-Interim data used in this paper were downloaded from the National Center for Atmospheric Research (NCAR) climate data website (http://www.ecmwf.int/research/era).

### 2.4. VTH Product From 1DVAR Approach

By constructing a pair of spectral indices (a so-called beta ratio) from observations made at 8.6, 11.0, 12.0, and 13.3 μm, background effects such as surface temperature, surface emissivity, and the ambient atmospheric conditions can be taken into account. A volcanic ash retrieval approach, which has served as the official operational volcanic ash algorithm for the National Oceanic and Atmospheric Administration's (NOAA) new-generation weather satellite GOES-R series, was developed based on beta ratios, forward simulations of IR radiances, and 1DVAR inverse method [5,29,47]. This algorithm (hereafter referred to as the 1DVAR approach) requires real-time or near-real-time (NRT) atmospheric profile data as input to drive simulations of clear and cloudy sky radiances, and it is more accurate under clear sky and single-layer cloudy sky conditions.

In this work, VTHs retrieved from SEVIRI radiance measurements using both the 1DVAR approach and the hybrid deep-learning-based algorithm, are compared and validated with CALIOP VTH product (see details in Section 4).

### 2.5. Data Preprocess and Quality Control

There are two main steps to preprocess all sources of data before constructing the hybrid retrieval model. In the first step, SEVIRI L1 data, CALIOP L2 cloud top data, the atmospheric profile data, and the VTH product with 1DVAR approach, from the eruptions, were collected and colocated. In the second step, a CALIOP overpass overlaying on the contemporary SEVIRI RGB false-color image, together with the volcanic ash mask using NOAA's volcanic ash detection algorithm [29], were analyzed

(see Figure 1 as an example). The points that fell into the cloud mask area along the contemporary CALIOP overpass were collected for further analysis. The corresponding CALIOP L2 cloud top heights of these points are considered as "true" VTH values. In this way, a total of 1900 colocated samples were collected from the Icelandic Eyjafjallajökull volcanic eruptions, from which 1500 samples were randomly selected for training while the other 400 samples were used for validation. In addition, a total of 1280 samples were collected from the eruptions of the Puyehue-Cordón Caulle volcanic complex (PCCVC), from which 1000 samples were randomly selected as training while the other 280 samples were used for validation.

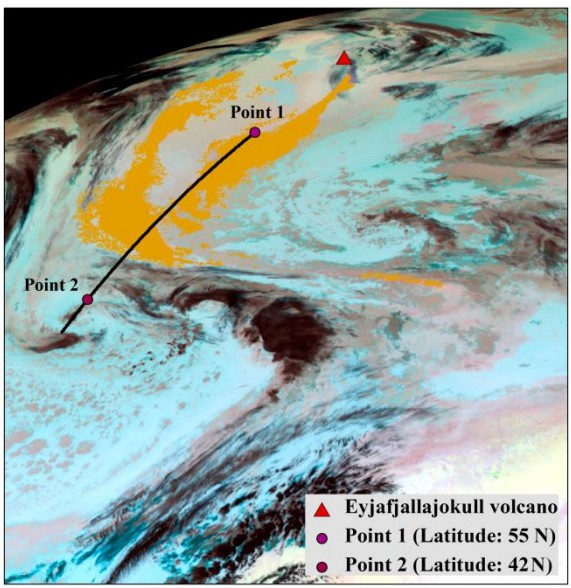

**Figure 1.** An example of Spinning Enhanced Visible and Infrared Imager (SEVIRI) RGB false-color image (red: 12-10.8μm, green: 10.8-8.7μm, and blue: 10.7μm) from 04:00 UTC on 8 May 2010 (Eyjafjallajökull eruption). The contemporary Cloud-Aerosol Lidar with Orthogonal Polarization (CALIOP) overpass (black line) and volcanic ash mask (yellow area) are overlaid.

## 3. Methodology

### 3.1. SDA-GA-LSSVR Model

Deep learning techniques developed for visual pattern and speech recognition have become more generalized, with improved transfer learning properties [48–50]. Deep learning techniques can automatically learn features from training data, and this learning can be used for feature extraction to provide highly effective solutions to pattern recognition problems. In this section, SDA is combined with the LSSVR and the GA to retrieve VTH from SEVIRI brightness temperatures (BT) measurements. In order to ensure that overfitting occurs, the k-fold cross-validation was used for training data. Figure 2 is the flowchart for the VTH algorithm in this study. The specific implementation steps are as follows:

1.　BT data from the SEVIRI's 6 IR channels, which are channel 4 (3.9 μm), 5 (6.25 μm), 7 (8.7 μm), 9 (10.8 μm), 10 (12.0 μm), and 11 (13.4 μm), VTH product derived from CALIOP observations, and atmospheric temperature vertical profile data from the ERA-Interim reanalysis [46] are used as the original features (samples). These data are preprocessed and further normalized to eliminate the effect of nonuniform dimension lengths for training the hybrid deep learning model.

2.　Based on the collected original features, SDA were performed through the following 3 steps: (1) To set the values of super-parameters such as the number of nodes in each layer of the SDA (learning rate = 0.1, input zero masked fraction = 0.5, activation function = "sigm"); (2) To perform a layer-by-layer pretraining to find the local optimum for each noise-reducing self-encoder parameter; and finally, (3) the pretrained noise-reducing self-encoder parameters for all layers are

formed into a neural network, and an unsupervised training is carried out. Through this step, the network is automatically optimized and the new features which highly represent the main characteristics of the original data were generated.

3. The new features extracted by the SDA were further passed to the least squares support vector machine, and the LSSVR model was established to estimate the VTH. In this process, the GA was used to optimize 2 key parameters for LSSVR, which are the regularization parameter *c* and the radial basis function *g*, respectively.

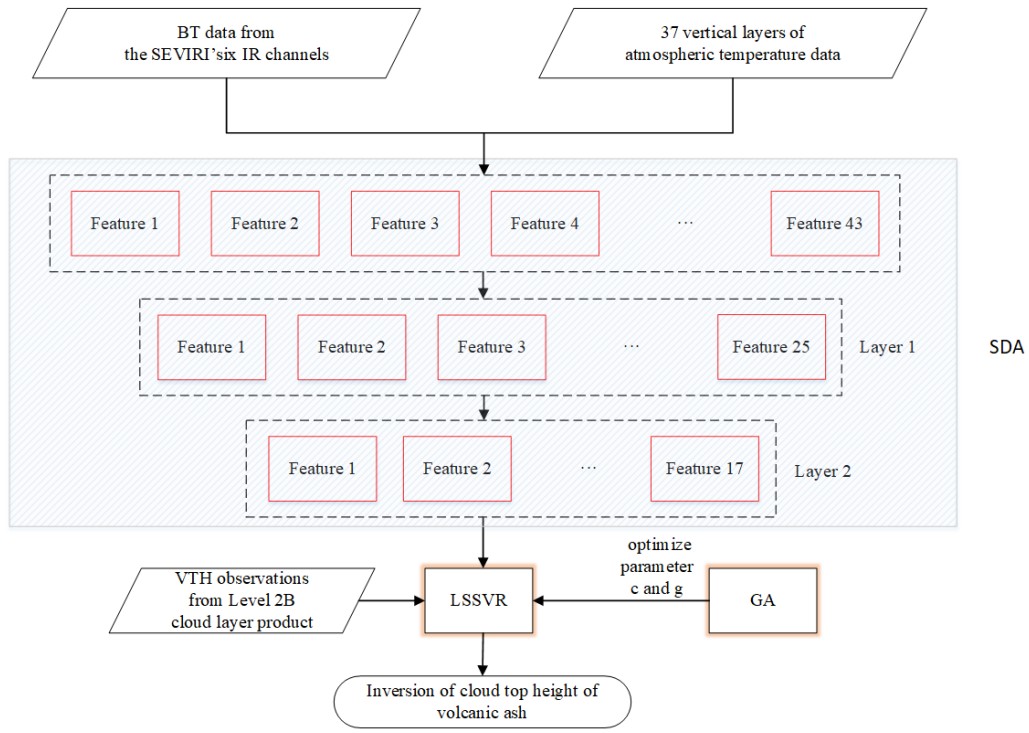

**Figure 2.** Flowchart of procedures in the Stacked Denoising AutoEncoder, Genetic Algorithm, and Least Squares Support Vector Regression (SDA-GA-LSSVR) model.

### 3.2. Evaluation of Retrieval Accuracy

Four evaluation indices were used to verify the accuracy of the VTH retrievals during validation: the bias, the mean absolute percentage error (MAPE), the standard deviation (STD), and the correlation coefficient (R). The formulae for the four evaluation indexes are defined by the following equations:

$$bias = \frac{1}{n} \sum_{i=1}^{n} \left| \hat{Y}_i - Y_i \right|, \tag{1}$$

$$MAPE = \frac{1}{n} \sum_{i=1}^{n} \left| \frac{\hat{Y}_i - Y_i}{Y_i} \right|, \tag{2}$$

$$R = \frac{\sum_{i=1}^{n} \left( \hat{Y}_i - \overline{\hat{Y}} \right) \left( Y_i - \overline{Y} \right)}{\sqrt{\sum_{i=1}^{n} \left( \hat{Y}_i - \overline{\hat{Y}} \right)^2 \sum_{i=1}^{n} \left( Y_i - \overline{Y} \right)^2}}, \tag{3}$$

$$STD = \sqrt{\frac{1}{n} \sum_{i=1}^{n} \left( Z_i - \mu \right)^2}, \tag{4}$$

where $n$ is the number of samples used for validation, $Y_i$ is the actual value of the $i$th VTH, $\overset{\wedge}{Y}_i$ is the retrieval value of the $i$th VTH, $Z_i$ is the value of $\overset{\wedge}{Y}_i - Y_i$, and $\mu$ is the average of $Z_i$.

## 4. Results and Validation

### 4.1. Eyjafjallajökull Eruption

To verify the VTH retrievals from the hybrid SDA-GA-LSSVR model, a series of eruptions over Iceland's Eyjafjallajökull volcano from April to May 2010 were selected as typical suitable cases. Back Propagation (BP), SDA, LSSVR, and GA-LSSVR were used as four benchmark models for comparisons with the hybrid SDA-GA-LSSVR model, and VTHs from CALIOP data in the independent samples were used as "truth" data for validation. The four evaluation indices mentioned above are used for inter-comparisons (Figure 3). Compared with the other four benchmark models, the hybrid SDA-GA-LSSVR model performs the best, with the lowest bias, STD, and MAPE, and the highest correlation with the CALIOP VTH product. Therefore, combining SDA, GA, and LSSVR increases the model performance.

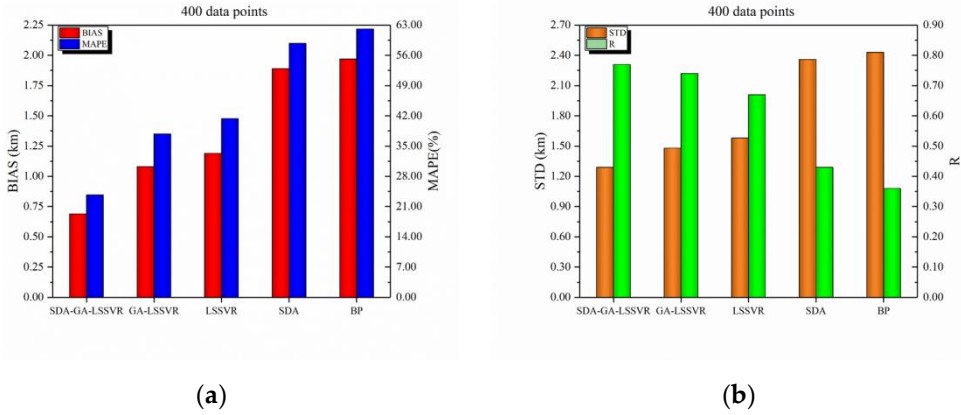

**Figure 3.** (**a**) shows bias and mean absolute percentage error (MAPE); (**b**) shows standard deviation (STD) and R calculated for volcanic ash cloud top height (VTH) retrievals using different models (Eyjafjallajökull eruption).

To analyze the influence of the SDA and the GA optimization on the proposed hybrid model, the following four additional comparisons were conducted. The first comparison (Comparison I) is intended to verify the superiority of SDA in terms of the retrieval accuracy. Results from the hybrid model are compared with those from the GA-LSSVR model. The second comparison (Comparison II) is intended to assess the contribution of the GA optimization to the LSSVR model by comparing retrievals from the GA-LSSVR model with those from the LSSVR model. The third and fourth comparisons (Comparisons III and IV) compare retrievals from SDA model with those from the LSSVR model and the BP model respectively. The comparison between the hybrid SDA-GA-LSSVR and the hybrid GA-LSSVR in Table 1 shows that the SDA-GA-LSSVR model achieves reduced bias, STD, and MAPE, but an increased R. This indicates the advantage of SDA on automatically extracting features through multiple hidden-layer learning structures without any further prior knowledge. In Comparison II, the GA optimization effectively improves the accuracy of retrievals from the LSSVR model and can be used to achieve a strong approximation capability. Comparison III shows that LSSVR is more effective for small sample learning, and the SDA is more effective for learning with large sample size. In Comparison IV, results from the SDA model and the BP model are compared because they both use the same error propagation mechanism in the fine-tuning. The main difference between the SDA and the BP is in the ways the weights and thresholds are set. In BP, the weights and thresholds are obtained after initialization, while in the SDA, weights and thresholds are known values, generated after the

pretraining encoding and decoding steps. Denoising the original data used in the SDA can reduce the correlation of the input data and increase the robustness of the retrieval.

**Table 1.** Four evaluation indexes of VTH retrievals from four different methods (Eyjafjallajökull eruption).

| Index (+ for Increase, − for Decrease) | Proportion (%) | | | |
|---|---|---|---|---|
| | SDA-GA-LSSVR VS. GA-LSSVR (I) | GA-LSSVR VS. LSSVR (II) | LSSVR VS. SDA (III) | SDA VS. BP (IV) |
| bias | −36.11 | −9.24 | −37.04 | −4.06 |
| STD | −12.84 | −6.33 | −33.05 | −2.88 |
| MAPE | −37.43 | −8.53 | −32.51 | −0.09 |
| R | 4.05 | 10.44 | 55.81 | 12.56 |

From the scatter plots of the VTHs from three machine learning models, along with the traditional 1-DVAR algorithm and the corresponding VTH product from CALIOP (marked as TRUE) (Figure 4), it can be seen that the hybrid SDA-GA-LSSVR model has the highest correlation with the "true" data (R = 0.77). The use of the GA-LSSVR yields a slightly reduced R of 0.74. When only the LSSVR was used, R is reduced further to 0.67. It is also worth noting that the three machine learning models generally perform better than the traditional 1-DVAR algorithm (GOES-R series algorithm) in this particular study.

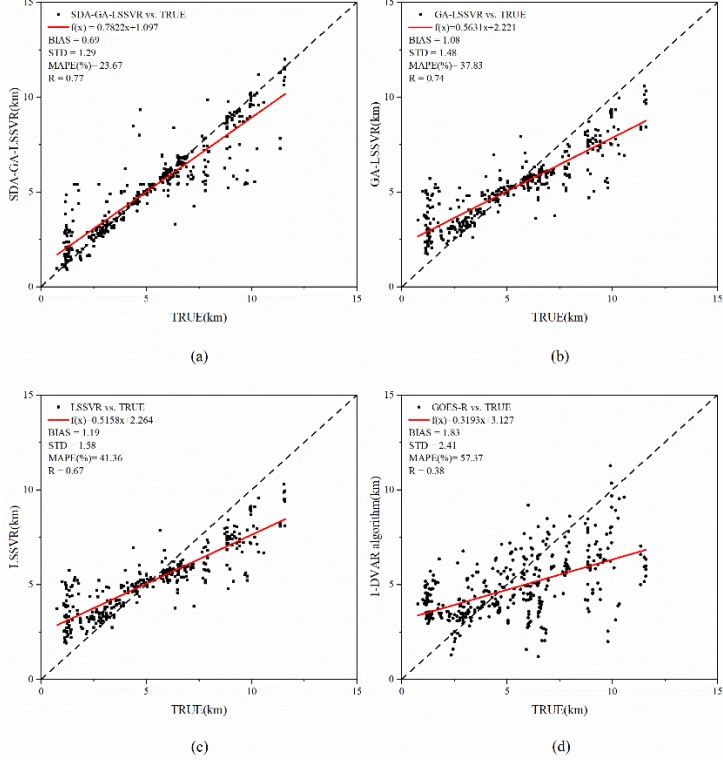

**Figure 4.** Scatterplots of the "true" VTHs from CALIOP versus the VTHs retrieved from (**a**) SDA-GA-LSSVR model, (**b**) GA-LSSVR model, (**c**) LSSVR model, and (**d**) one-dimensional variational (1DVAR) approach (Eyjafjallajökull eruption).

### 4.2. Puyehue-Cordón Caulle Eruption

In the Eyjafjallajökull cases, the meteorological background to the ash was relatively simple and the influence of meteorological cloud was weak. However, in many other cases, volcanic ash is mixed with meteorological cloud or consists of more than one vertical layer, which makes the retrieval of VTH more complicated [7]. To verify the performance of the hybrid SDA-GA-LSSVR model in a more complex meteorological situation, VTHs retrievals from a series of eruptions over the Puyehue-Cordón Caulle volcanic complex (PCCVC) in Chilean Andes in June 2011 are evaluated. The four evaluation indices, bias, STD, MAPE, and R, are shown in Table 2. Compared with the Eyjafjallajökull case, the bias, STD, and MAPE from the hybrid SDA-GA-LSSVR model increase slightly (Figure 5), which is probably attributed to the increased complexity of the atmospheric background. In general, the hybrid SDA-GA-LSSVR model still provides the highest accuracy, with the lowest bias, STD, and MAPE, but the highest correlation with CALIOP observations in this study, when compared with the other three machine learning models.

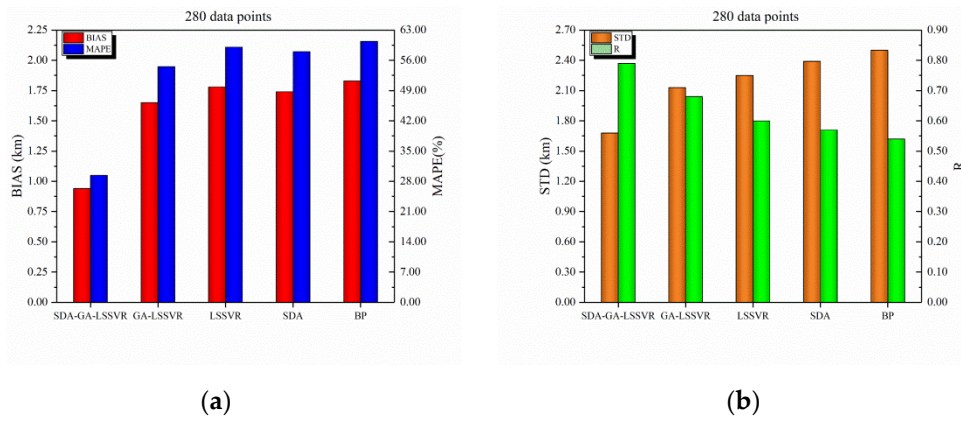

(**a**)  (**b**)

**Figure 5.** (**a**) shows bias and MAPE; (**b**) shows STD, and R calculated for VTH retrievals using different models (Puyehue-Cordón Caulle eruption).

The scatter plots of VTH retrievals from the four machine learning models, and the CALIOP VTH product show that the hybrid SDA-GA-LSSVR model has the highest correlation with the "true" VTHs (R = 0.79) (Figure 6). When the GA-LSSVR was used, R was decreased to 0.68. When only the LSSVR was used, R was further decreased to 0.60. It is also worth noting that the four machine learning models also perform better than the traditional 1DVAR retrieval method under these more complicated meteorological conditions in this particular study.

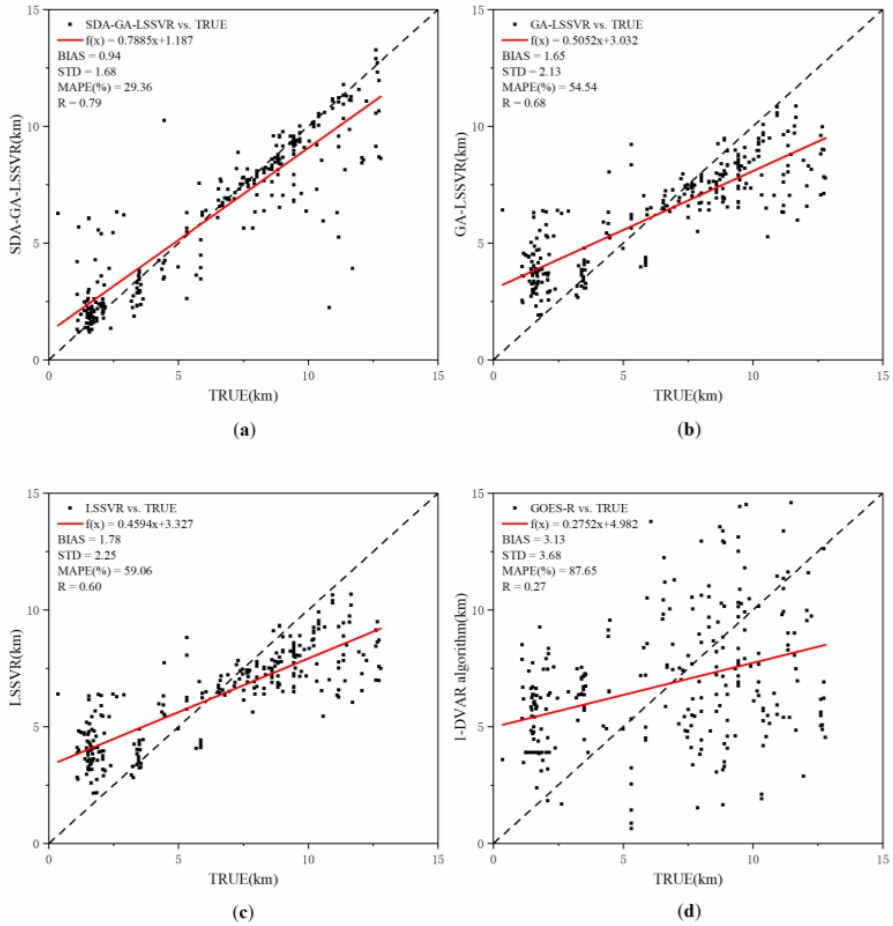

**Figure 6.** Scatterplots of the "true" VTHs from CALIOP versus the VTHs retrieved from (**a**) SDA-GA-LSSVR model, (**b**) GA-LSSVR model, (**c**) LSSVR model, and (**d**) 1DVAR approach (Puyehue-Cordón Caulle eruption).

Table 2 presents paired comparisons of the four machine learning models. The SDA model shows an improved estimation ability compared with the BP, as bias, STD, and MAPE are all reduced and R is increased (Table 2, Comparison IV). For Comparison III, where the SDA is compared to the LSSVR, both bias and MAPE are increased while STD is decreased. It is therefore difficult to conclude that the LSSVR results have improved accuracy over the SDA (Comparison III). If the GA is used with the LSSVR, the accuracy is increased (Comparison II), suggesting that through the hybrid use of the GA and the LSSVR, the retrieval accuracy could be further improved. If the SDA is further used with the GA-LSSVR (Comparison I), the bias is increased sharply from 7.3% to 43.03%, the STD is increased from 5.33% to 21.13%, and the MAPE is increased from 7.65% to 46.17%.

**Table 2.** The four evaluation indexes from the four different methods (Puyehue-Cordón Caulle eruption).

| Index (+ for Increase, − for Decrease) | Proportion (%) | | | |
|---|---|---|---|---|
| | SDA-GA-LSSVR VS. GA-LSSVR (I) | GA-LSSVR VS. LSSVR (II) | LSSVR VS. SDA (III) | SDA VS. BP (IV) |
| bias | −43.03 | −7.3 | 2.29 | −4.92 |
| STD | −21.13 | −5.33 | −5.86 | −4.4 |
| MAPE | −46.17 | −7.65 | 1.83 | −3.94 |
| R | 16.18 | 7.54 | 9.67 | 6.68 |

Different from the Eyjafjallajökull cases, the meteorological situations are more complicated for the Puyehue-Cordón Caulle eruption cases. There always is more than one layer of cloud in the vertical scale (such as Puyehue-Cordón Caulle case at 04:00 UTC, 16 June 2011, when there is a meteorological cloud underlying volcanic ash cloud, see Figure 7) [7]. When compared with results from Eyjafjallajökull cases, the hybrid SDA-GA-LSSVR model not only performs well under a relatively simple meteorological background but is also robust under more complex meteorological conditions, as seen in the Puyehue-Cordón Caulle cases (Figure 6), which indicates that the hybrid SDA-GA-LSSVR model has the ability to simulate complicated nonlinear relationship between IR radiances and volcanic ash cloud parameters through deep learning. In contrast, the traditional 1DVAR inverse method has the relatively lowest correlation with the "true" data in Puyehue-Cordón Caulle cases (R = 0.27) (Figure 6d). Currently, most forward RTM models used in the 1DVAR inverse method have been limited to clear skies or single-layer cloud situations. In the case of vertically distributed multilayer clouds, forward RTM models cannot accurately simulate the IR radiative transfer process, potentially resulting in large VTH retrieval uncertainties.

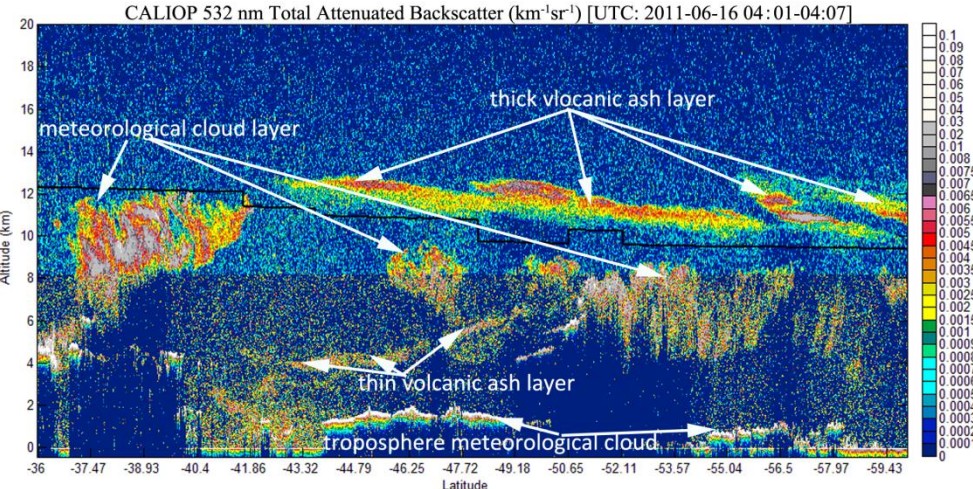

**Figure 7.** A CALIOP 532 nm total attenuated backscatter cross-section from 04:01 UTC to 04:07 UTC on 16 June 2011 (Puyehue-Cordón Caulle eruption). The solid black line denotes the tropopause. (This figure is from reference [7]).

## 5. Sensitivity to Atmosphere Temperature Vertical Profile Data

The atmosphere temperature vertical profile is a key driver for the radiative transfer simulations required for forward modeling. Regardless of whether the traditional 1DVAR inverse method or the statistical estimation method is used, retrievals are inevitably sensitive to meteorological conditions [5,7]. In this section, a sensitivity analysis was performed to test the sensitivity of the hybrid SDA-GA-LSSVR model to the atmospheric temperature by adding temperature vertical profile data to the training samples. Such atmospheric profiles can be derived in NRT from numerical weather prediction (NWP) model-based short-range forecasts or from the advanced geostationary imager measurements [25,51]. The inclusion of temperature profile data in the training samples resulted in reduced bias, STD, and MAPE and increased R. Table 3 demonstrates that adding atmospheric temperature vertical profiles to the training samples can further improve the accuracy of VTH retrievals.

**Table 3.** Impact of atmospheric temperature on VTH retrievals.

| (+ for Increase, − for Decrease) | | bias (km) | STD (km) | MAPE (%) | R |
|---|---|---|---|---|---|
| **Eyjafjallajökull** | **Adding profile data** | 0.69 | 1.29 | 23.67 | 0.77 |
| | **No profile data** | 1.16 | 1.78 | 35.82 | 0.59 |
| | **Adding VS. No** | −40.51 | −27.53 | −33.92 | +29.51 |
| **Puyehue-Cordón Caulle** | **Adding profile data** | 0.94 | 1.68 | 29.36 | 0.79 |
| | **No profile data** | 1.05 | 1.80 | 32.11 | 0.76 |
| | **Adding VS. No** | −10.48 | −6.67 | −8.56 | +3.95 |

## 6. Sensitivity of VTH Retrieval to Feature Selection with the SDA Model

A SEVIRI image acquired at 04:00 UTC on May 08, 2010 was used to assess the sensitivity of VTH retrieval to feature selection with the SDA model. Pattern recognition performance can be greatly improved by using the SDA to extract features. Figure 8 shows grayscale images of the radiances in each of the original six IR channels. The outline of the volcanic ash cloud can only be clearly seen in the channel 7 (8.7 μm), 9 (10.8 μm), and 10 (12.0 μm). It is difficult to detect the volcanic ash cloud information directly in the images from the other three IR channels. It is also difficult to infer any height variation from the images. The original grayscale images were compared with new features extracted from the SDA model (Figure 9). Figure 9 shows that most information about the volcanic ash cloud is contained in features 2 and 3, and there is very limited information in features 1 and 4 since channel decorrelation is performed by the SDA model.

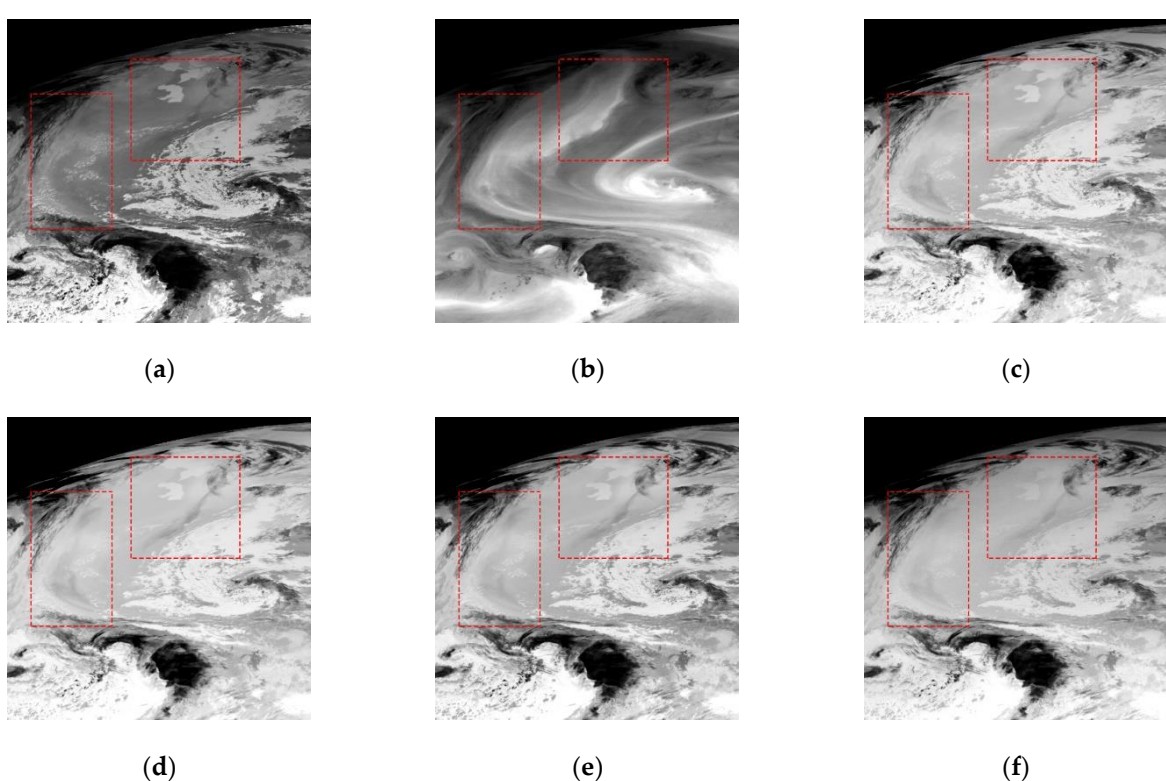

**Figure 8.** Brightness temperatures (BTs) of original six grayscale IR channels from SEVIRI images acquired at 04:00 UTC on 8 May 2010. (**a**): SEVIRI channel 4; (**b**): SEVIRI channel 5; (**c**): SEVIRI channel 7; (**d**): SEVIRI channel 9; (**e**): SEVIRI channel 10; (**f**): SEVIRI channel 11 (The red dotted box indicates the location of the volcanic ash cloud).

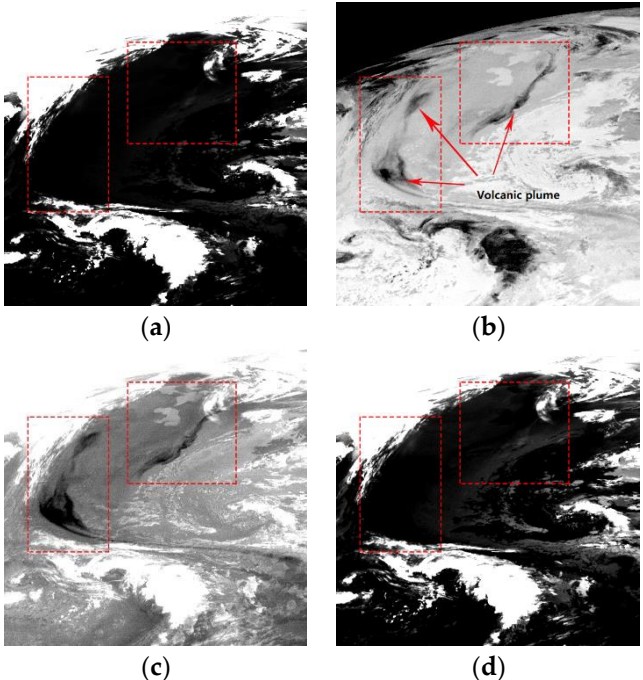

**Figure 9.** Grayscale images of four features identified by the SDA from SEVIRI images acquired at 04:00 UTC on 8 May 2010. (**a**): Feature 1; (**b**): Feature 2; (**c**): Feature 3; (**d**): Feature 4 (The red dotted box indicates the location of the volcanic ash cloud).

The corresponding ash area in features 2 and 3 appears darker in the grayscale image. This shows that through using the SDA model, information useful in estimating the VTH can be enhanced, which demonstrates the advantage of the SDA deep learning method on retrieving VTH.

The performance of a deep learning model depends on the choice of hyperparameters. For the SDA model, the choice of hyper-parameters is particularly important. The SDA's hyperparameters include the cycle count, the learning rate, a momentum term, and a noise reduction factor. In general, there are three ways to set the hyperparameters: random searching, grid searching, and manually setting. In Section 4, an automatic grid searching method was used to set hyperparameters in the SDA model. To test the sensitivity of VTH retrieval to the hyperparameters with the SDA model, the choice of hyperparameters was changed from grid searching to manually setting. By changing the value of the layer-by-layer pretraining learning rate ([0.1:0.1:1.0]), ten parameter sets are created for the experiments. A loss function (Equation (5) below) was used to measure the ability of the SDA model to reconstruct the original feature matrix for each parameter set. The lower the value is, the better ability the reconfiguration has.

$$LOSS = \frac{1}{2m} \sum_{i=1}^{m} \sum_{j=1}^{n} \left(A_{ij} - B_{ij}\right)^2, \tag{5}$$

In Equation (5), $A_{ij}$ and $B_{ij}$ are the original matrix and the reconstruction matrix, respectively. Parameters $m$ and $n$ are the number of rows and columns of the matrix, respectively.

Table 4 and Figure 10 show the relationship between the different parameter sets used for the SDA reconstruction of the original features and the final MAPE of the derived VTHs. The LOSS varies with the learning rate. The highest loss function value is 0.0921 and the lowest is 0.0475, which indicates that the choice of hyperparameters directly influences the learning ability of the SDA. The MAPE also varies with LOSS. When the LOSS is 0.0832, the MAPE value peaks at 0.4741. When the LOSS is 0.0502, the MAPE reaches its lowest value of 0.2096. Figure 10 shows that there is a significant positive correlation between LOSS and MAPE (R = 0.62). These results suggest that the selection of features

directly affects the accuracy of the final retrieval. In addition, the results obtained using the manual searching method, rather than using the grid searching method, for selecting the hyperparameters, are not stable. Therefore, implementing the grid searching method improves the accuracy and stability of the model.

**Table 4.** Influence of feature selection on inversion results.

| LOSS | MAPE | Pretraining Learning Rate |
| --- | --- | --- |
| 0.0502 | 0.2096 | 0.1 |
| 0.0921 | 0.3644 | 0.2 |
| 0.0683 | 0.1906 | 0.3 |
| 0.0805 | 0.4285 | 0.4 |
| 0.0792 | 0.3209 | 0.5 |
| 0.0832 | 0.4741 | 0.6 |
| 0.0508 | 0.2477 | 0.7 |
| 0.0537 | 0.3071 | 0.8 |
| 0.0513 | 0.2173 | 0.9 |
| 0.0475 | 0.217 | 1 |

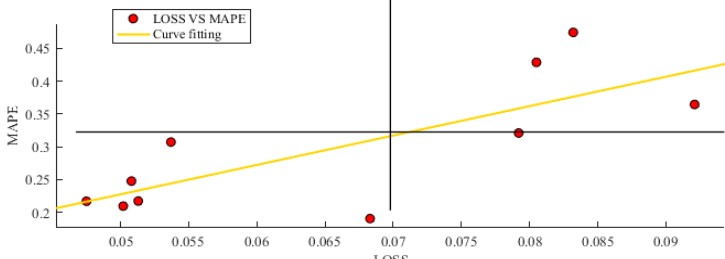

**Figure 10.** Scatter plot between LOSS and MAPE.

## 7. Uncertainty Analysis

In this study, four types of data were used together to develop a hybrid SDA-GA-LSSVR deep learning model for VTH retrieval, and each data source may introduce uncertainties on the final retrievals.

First, in the data preprocessing stage, although the cloud mask and the RGB false-color image are used together to further distinguish meteorological and volcanic ash clouds, uncertainties may still exist under certain situations. For example, when the volcanic ash cloud is very thin with a meteorological cloud underlying (Figure 11), the CALIOP L2 cloud top height data may reflect the height of the underlying meteorological cloud instead of the VTH. This may lead to underestimation of VTHs. In addition, the volcanic ash mask used in this study is derived from NOAA's volcanic ash detection algorithm [29], and the algorithm may produce some false alarms or missing ash/dust clouds when it is applied in global scale [5]. Further investigation is needed on better distinguishing volcanic ash clouds and meteorological clouds.

Second, the horizontal and vertical variability of the atmospheric temperature may not be captured well by very coarse-resolution ERA-Interim, especially near the surface. In the future, the input ERA-Interim data could be replaced by data with higher horizontal or vertical resolution, such as ERA5 or GNSS Radio Occultation data [52].

Third, the process of collocating different sources of data such as SEVIRI L1 data, CALIOP L2 cloud top data, atmospheric vertical profile data, etc. may also introduce additional uncertainties. The spatial resolution of SEVIRI L1 IR data and cloud mask data is 3 km, which may reach to 10 km or greater in high latitude. While the horizontal resolution of CALIOP L2 cloud top data used in this study is 5 km. The horizontal resolution of atmosphere vertical profile data is 80 km. Large VTH retrieval errors may occur when the GEO satellite viewing angle is large.

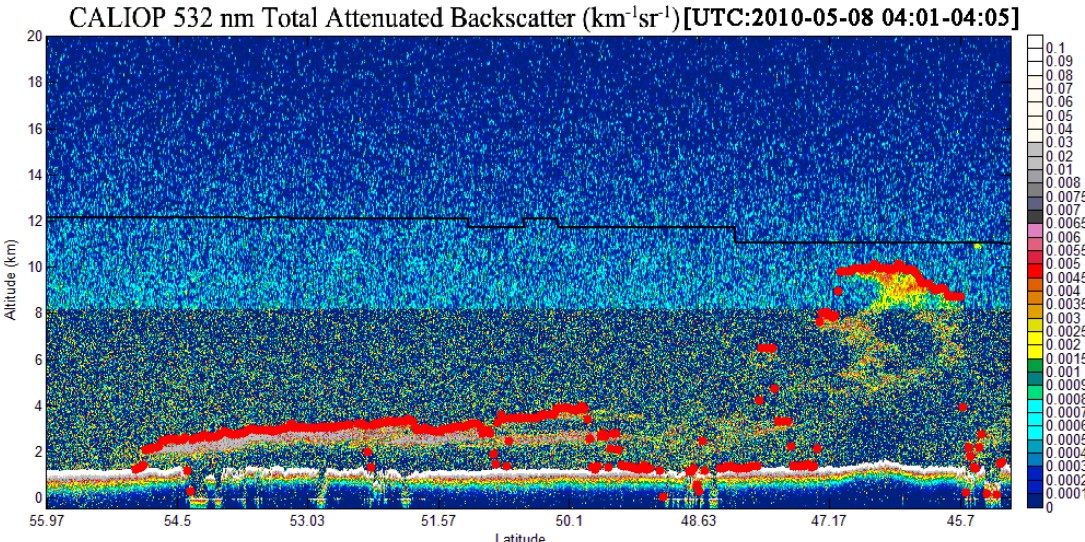

**Figure 11.** An example of CALIOP 532 nm total attenuated backscatter cross-section from 04:01:59 UTC–04:05:04 UTC on 8 May 2010 (Eyjafjallajökull eruption). The contemporary CALIOP L2 cloud heights products (red dots) are overlaid.

## 8. Conclusions

In this study, BT data from the SEVIRI onboard the Meteosat-8 satellite platform and the VTH product from CALIOP were used to construct a sample dataset for developing a deep-learning-based statistical retrieval model. A deep-learning-based method through hybrid use of the SDA and the LSSVR, optimized by the GA, has been developed to establish a nonlinear relationship between SEVIRI IR measurements and VTH. The following conclusions can be drawn from independent validation and analysis of the results:

1. By comparing the retrievals obtained from the hybrid SDA-GA-LSSVR, the GA-LSSVR, the LSSVR, the SDA, and the BP models for two typical cases, it is found that the GA optimization algorithm can effectively improve the approximation of the LSSVR and improve the accuracy of VTH retrievals. For small samples, the LSSVR is a novel learning method with a solid theoretical basis. The SDA performs better for larger samples size. The SDA model and the BP model are compared because they both use the same mechanism for fine-tuning. The use of the SDA to denoise the satellite measurements can reduce correlation among the input data and increase the robustness of the retrieval. The hybrid uses of the SDA, the GA, and the LSSVR achieve the most accurate VTH retrievals, with the smallest error and highest correlation with the "true" data (CALIOP measurements).

2. Since the hybrid SDA-GA-LSSVR model has the ability to simulate the complicated nonlinear relationship between IR radiances and the volcanic ash cloud parameters through deep learning, it not only performs well under a relatively simple meteorological background but also is robust under more complex meteorological conditions, as seen in the Puyehue-Cordón Caulle cases (R = 0.79).

3. Using the hybrid SDA-GA-LSSVR retrieval algorithm, the nonlinear relationship between IR channel observations and VTH can be well established. However, due to the uncertainties that are attributed to different eruption times, atmospheric conditions, surface conditions, and satellite observation angles, it is difficult to fully capture temporal and spatial changes in VTH retrievals using only satellite IR observations as input, especially in the case of complex meteorological conditions. Adding atmospheric temperature vertical profiles to the training samples results in significant reduced bias, STD, and MAPE but increased R; the R is increased by 3.95% for the Eyjafjallajökull cases and 29.51% for the Puyehue-Cordón Caulle cases. These results demonstrate

that adding atmospheric temperature vertical profile information to training samples can further improve the VTH retrievals, while the moisture profiles do provide a little impact since moisture mostly resides in low troposphere and has little radiative effect on the IR channels used for VTH retrieval (results now shown).

4.  The IR channel observations are spectrally correlated. The SDA can automatically extract complex features due to its multi-hidden-layer structure. Therefore, the SDA is capable of enhancing volcanic ash information from satellite IR measurements. In addition, the choice of hyperparameters directly influences the learning ability of the SDA and the final retrievals.

Since only the eruptions over Iceland's Eyjafjallajökull volcano from April to May 2010, and the eruptions over the Puyehue-Cordón Caulle volcanic complex in Chilean Andes in June 2011 were considered in constructing training samples, more work is needed to include more volcanic eruption cases in future for training so that the samples contain a variety of volcanic ash spectral and mineral properties.

China's new generation of GEO meteorological satellites, Fengyun 4A (FY-4A) [53,54], has finished in-orbit tests, and data were released to users worldwide in May 2018. The primary instrument AGRI (the Advanced Geosynchronous Radiation Imager) onboard FY-4A has similar infrared channels to those included in instruments onboard the Geostationary Operational Environmental Satellite (GOES-R series, NOAA), and Himawari-8/-9 (JMA). The hybrid deep learning method has the potential to be used to exploit imager data from this new generation of international GEO weather satellites and to provide global VTH products. It is also applicable to the measurements from the hyperspectral IR sounders [55], especially the hyperspectral IR sounders from GEO orbit [56,57].

**Author Contributions:** Conceptualization, L.Z. and J.L.; Methodology, W.Z.; Project administration, L.Z. and J.L.; Supervision, J.L. and L.Z.; Writing—original draft, W.Z.; Writing—review & editing, J.L., L.Z. and W.Z.; Resources, J.L., L.Z. and H.S. All authors have read and agreed to the published version of the manuscript.

**Funding:** This research was funded by the National Key Research and Development Program, grant number 2018YFA0605502, the National Natural Science Foundation of China, grant number 41871263, and NOAA GOES-R Science Program at CIMSS NA15NES4320001.

**Acknowledgments:** The authors thank Michael Pavolonis of the National Oceanic and Atmospheric Administration Center, who provided GOES-R volcanic ash products for the two cases (Eyjafjallajökull eruption and Puyehue-Cordón Caulle eruption) so that we could better reference the volcanic ash cloud area and compare the deep learning inversion results with a one-dimensional variational inversion method. We also would like to thank two anonymous reviewers for their valuable suggestions and comments, which helped the authors think deeply about some theoretical and technical issues and significantly improve the manuscript.

**Conflicts of Interest:** The authors declare no conflict of interest.

## Appendix A. Algorithm Principle

### *Appendix A.1. Stack Noise Reduction Encoder*

The autoencoder is an unsupervised learning model, first proposed by Rumelhart in 1986 [58]. It is a kind of neural network and is often used in deep learning techniques. The network consists of two parts: an encoder $f(x)$ and a decoder $g(x)$. It is assumed that the automatic encoder inputs an n-dimensional signal $x$ to the input layer. In the intermediate layer, the signal becomes h as follows:

$$h = f(Wx + b), \tag{A1}$$

where W is the connection weight for the input layer to the middle layer, and b is the offset of the middle layer. This processing of x to give h is the encoding process. The signal h is decoded by the decoder, and output to the output layer of the neuron. In the output layer, the signal becomes r as follows:

$$r = g(W\prime x + b'), \tag{A2}$$

where W′ is the connection weight for the middle layer to the output layer, $b'$ is the offset of the output layer, and the processing of h to give r is the decoding process. The autoencoder is trained to copy the input so that h captures useful characteristics from x and is independent of the decoder output. One way to obtain useful features from a self-encoder is to limit the dimension of h to be smaller than x. An autoencoder whose coding dimension is smaller than the input dimension is called an undercomplete autoencoder. Learning an undercomplete representation forces the encoder to capture the most significant features in the data.

The training can be described as the minimization of a loss function:

$$L(x, g(f(x))), \tag{A3}$$

where L is a loss function used to minimize the difference between $g(f(x))$ and x. When the decoder is linear and L is the root mean square error, the undercomplete encoder will generate a subspace equivalent to a Principal Component Analysis (PCA). In contrast to PCA, the autoencoder learns the principal subspace of the training data while training to perform the copy task. Therefore, an autoencoder with a nonlinear encoder function $f(x)$ and a nonlinear decoder function $g(x)$ represents a powerful nonlinear extension of PCA and can be used to capture multimodal aspects of the input data.

The SDA is an autoencoder that handles training data which includes noise. It was first proposed by Bengio in 2007 [59]. To avoid problems associated with overfitting, noise is added to the input data, making the learned encoder more robust and enhancing the potential of the model for generalized applications. The principle behind the noise reduction encoder is similar to that of the autoencoder. In the input layer, noise is randomly added to the signal for encoding, using a fixed signal-to-noise ratio.

For a single-layer autoencoder with a hidden layer, a backpropagation algorithm is usually used for training, and this can be effective. However, if a backpropagation algorithm is applied to a network with multiple hidden layers, then the error is likely to become extremely small after the first few layers and the training will become invalid. Although more advanced backpropagation methods have addressed this problem, the additional problem of a slow learning speed remains, particularly when the amount of training data is limited. By pretraining each layer as a simple self-decoder and then stacking it, training efficiency is greatly improved. The implementation steps for the two-layer stack noise reduction encoder are as follows.

1.  Take a training sample $x$ from the training data.
2.  Set the noise reduction ratio k, denoise $x$ to obtain new input information $\widetilde{x}$, where nk data in $\widetilde{x}$ is 0.
3.  Replace the input information with $\widetilde{x}$, estimate the reconstructed distribution of the self-encoder, and re-enter the training.
4.  Using the first three steps to denoise the first Denoising Autoencoder (DA) unit, use the hidden layer of the first DA unit as input to the second DA unit, and then denoise again to extract the hidden layer as the feature output.

For this study, the sample consists of six dimensions, corresponding to brightness temperatures in six SEVIRI channels.

*Appendix A.2. Stack Noise Reduction Encoder Least Squares Support Vector Regression Method Based on a Genetic Algorithm*

The support vector machine is a machine learning model proposed by Suykens in the 1990s and is important in the field of remote sensing [60]. In contrast to traditional linear regression methods, support vector machines use complex optimization techniques to classify features according to nonlinear relationships within the training data. This paper uses a computationally efficient least squares support vector machine with a fast solution speed to construct nonlinear relationships between the spectral characteristics and heights of volcanic ash clouds. The set of data points found after

the SDA calculation is as follows: $A = \{(u_i, y_i) | i = 1, 2, \ldots n\}$, where $u_i \in R^m$ and $m$ is the number of dimensions calculated by the SDA from the brightness temperatures in the six channels. The next step is to use a nonlinear mapping $\varphi: R^m \rightarrow H$ to map $x$ to the high-dimensional feature space $H$. This gives a regression function describing the input and output:

$$y(u) = w\varphi(u) + b, \tag{A4}$$

where $w$ is a weight, b is a bias, and $\varphi(u)$ is a nonlinear function.

The LSSVR model follows from solution of the following optimization problem:

$$\text{Minimize}: \ J(w, a) = \frac{1}{2}\|w\| + \frac{c}{2}\sum_{i=1}^{n} \rho_i^2, \tag{A5}$$

$$\text{Subject}: \ y_i = w'v(u_i) + b + \rho_i, \tag{A6}$$

where $c$ is a regularization parameter and $\rho_i$ is a relaxation variable. To solve the above optimization problem, the constraint optimization problem becomes an unconstrained optimization problem, and the Lagrangian function is introduced:

$$L(w, a, b, \rho) = J(w, a) - \sum_{i=1}^{n} a_i(w'v(u_i) + b + \rho_i - y_i), \tag{A7}$$

where $a_i$ is a Lagrangian multiplier, $\sum_{j=1}^{n} a_j = 0$. After canceling $\rho_i$ and $w$, the following linear equations are obtained:

$$\sum_{j=1}^{n} a_j(\varphi(u_i), \varphi(u_j)) + b + a_i/c = y_i, \tag{A8}$$

Using the kernel function $k(u_i, u_j) = (\varphi(u_i), \varphi(u_j))$, (A3) is converted to:

$$\sum_{j=1}^{n} a_j k(u_i, u_j) + b + a_i/c = y_i, \tag{A9}$$

If the number of features is much smaller than the number of samples, Radial Basis Function (RBF) can be used as the kernel function and the regression function becomes:

$$y(u) = \sum_{i=1}^{n} a_i exp(-\frac{\|U - U_i\|^2}{2g^2}) + b, \tag{A10}$$

When dealing with regression problems, LSSVR requires the regularization parameter $c$ and the radial basis function $g$ of the function to be set. The choice of values for these parameters has an important influence on the effectiveness of the regression. The values are usually based on an empirical method, which makes the results uncertain. In this study, the genetic algorithm is used to intelligently search the parameters in order to extract the optimal parameters to improve the effectiveness of the ash cloud height derivation.

GA is often used to solve optimal problems under multiple constraints. It is a random search method derived from evolutionary biology (the genetic mechanism that underlies the "survival of the fittest" concept). It was first proposed by Professor J. Holland in the United States in 1973 [61], who used it to simulate natural evolution processes. The genetic algorithm is able to quickly search the entire solution space to find the global optimal solution, avoiding local minima in the optimization to ensure that the calculated solution is not just a local optimal solution.

In this paper, the genetic algorithm is used to obtain the optimal parameters required by the LSSVR prediction model. The optimization process is as follows:

1.  Set the largest evolutionary algebra and randomly generate the number of individuals as the initial state (initialization).
2.  Calculate the fitness of each individual in the initial population (calculation of fitness).
3.  Determine recombinant or crossed individuals and subindividuals according to the fitness obtained above (select).
4.  Generate new individuals by mating the information from the population with the father (cross).

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
