# Peer review of "Retrieving Volcanic Ash Top Height through Combined Polar Orbit Active and Geostationary Passive Remote Sensing Data"

_remotesensing, doi:10.3390/rs12060953_

Round 1
Reviewer 1 Report
The paper shows the capability of a deep learning algorithm to retrieve the volcanic ash cloud top heights focusing on 2 specific eruptions: Eyjafjallajökull 2010 and Puyehue-Cordón Caulle 2011. The authors used different satellite data (SEVIRI, GOES, CALIPSO) and ERA-Interim reanalyses as inputs and target to train and test the algorithm. The results show a good improvement on the retrieval of volcanic cloud top height due to the use of their “combined” algorithm (SDA-GA-LSSVR) and it looks promising. However, from my point of view, several parts of the paper must be improved and part of the work clarified. The introduction is not complete and does not fully represent the state of the art. It is not fully clear what data the authors used for the analysis and a new section should be added to explain the uncertainties taken in account. When working with remote sensing data, the uncertainties are always an issue. Here the authors rely on 3 different satellites data (and algorithms) each of them introducing uncertainties on the estimations. The paper lacks a section explaining all the uncertainties which should be considered such as uncertainty on volcanic cloud top height estimation from CALIOP, uncertainty on ash cloud detection from GOES-R, uncertainty on co-locations and so on … In addition, they also use ERA-Interim data with just 37 vertical layers for studying clouds which are usually very thin when move away from the volcano. A section explaining all the uncertainties and the possible impact on the algorithm is needed.
English and grammar must be improved.
All the figure captions must contain more details and must describe what is shown.
Due to the related topic I suggest to submit it to the Special Issue “Convective and Volcanic Clouds (CVC)”.
I report below the details for each section.
Introduction: not complete. There are several different techniques to detect the VTH not mentioned in this paper, which are strictly connected to the manuscript for different reasons (because using CALIOP, deep learning algorithms, SEVIRI, …), I suggest to cite all of them and to deepen the bibliography with other works for making the introduction fully comprehensive and to show the status of the art on this topic. I also suggest to refer to some of those papers on other sections of the manuscript.
Prata, A. J., & Grant, I. F. (2001). Determination of mass loadings and plume heights of volcanic ash clouds from satellite data. CSIRO Atmospheric Research.
Marzano, F. S., Barbieri, S., Vulpiani, G., & Rose, W. I. (2006). Volcanic ash cloud retrieval by ground-based microwave weather radar. IEEE transactions on geoscience and remote sensing, 44(11), 3235-3246.
Prata, A. J., & Kerkmann, J. (2007). Simultaneous retrieval of volcanic ash and SO2 using MSG‐SEVIRI measurements. Geophysical Research Letters, 34(5).
Corradini, S.; Merucci, L.; Prata, A.J.; Piscini, A. Volcanic ash and SO2 in the 2008 Kasatochi eruption:
Retrievals comparison from different IR satellite sensors. J. Geophys. Res. Atmos. 2010, 115, D00L21.
Corradini, S., Merucci, L., Prata, A. J., & Piscini, A. (2010). Volcanic ash and SO2 in the 2008 Kasatochi eruption: Retrievals comparison from different IR satellite sensors. Journal of Geophysical Research: Atmospheres, 115(D2).
Picchiani, M., Chini, M., Corradini, S., Merucci, L., Sellitto, P., Del Frate, F., & Stramondo, S. (2011). Volcanic ash detection and retrievals using MODIS data by means of. Atmospheric measurement techniques.
Zakšek, K., Hort, M., Zaletelj, J., & Langmann, B. (2013). Monitoring volcanic ash cloud top height through simultaneous retrieval of optical data from polar orbiting and geostationary satellites. Atmospheric Chemistry & Physics, 13(5).
Piscini, A., Picchiani, M., Chini, M., Corradini, S., Merucci, L., Del Frate, F., & Stramondo, S. (2014). A neural network approach for the simultaneous retrieval of volcanic ash parameters and SO2 using MODIS data. Atmospheric Measurement Techniques.
Merucci, L., Zakšek, K., Carboni, E., & Corradini, S. (2016). Stereoscopic estimation of volcanic ash cloud-top height from two geostationary satellites. Remote Sensing, 8(3), 206.
Corradini, S., Montopoli, M., Guerrieri, L., Ricci, M., Scollo, S., Merucci, L., ... & Grainger, R. G. (2016). A multi-sensor approach for volcanic ash cloud retrieval and eruption characterization: The 23 November 2013 Etna lava fountain. Remote Sensing, 8(1), 58.
Cigala, V., Biondi, R., Prata, A. J., Steiner, A. K., Kirchengast, G., & Brenot, H. (2019). GNSS Radio Occultation Advances the Monitoring of Volcanic Clouds: The Case of the 2008 Kasatochi Eruption. Remote Sensing, 11(19), 2199.
Section 2.2 CALIOP Data
Line 159 “the CALIOP Level 2B cloud layer product 159 is used to derive VTH” did the author estimate the VTH or they just read the values reported in the CALIOP cloud layer product? What CALIOP data the authors used? Also the horizontal resolution of these products can be different, since cloud layers are available with 333m, 1km and 5km resolution, which one they used? How the author could discriminate meteorological clouds from volcanic clouds when they were mixed?
Section 2.3 Atmospheric profile data
The use of ERA-Interim with only 37 layers could miss important features. Why the authors used just the temperature profiles and not other atmospheric parameters?
Just a suggestion for future works or for the revision of this paper: the ERA-Interim could be replaced by the use of higher vertical resolution profiles from GNSS Radio Occultations (e.g. Biondi et al., 2017, mentioned before)
I do not understand from the manuscript if the authors studied just the plume during the eruption (above the volcano) or they tracked the clouds (which I guess). In this second case how they could distinguish the ash from the meteorological clouds when the clouds were outside of the GOES-R coverage? It could be useful and interesting to see a map with all the co-locations.
Section 3.1
It is not clear to me if and how the authors optimized the machine learning models, did they try to understand what is the sensitivity of the algorithm to each single SEVIRI channel or temperature layer in the VTH estimation? Are all the parameters necessary?
Section 4.1
Figure 3 and line 258. I’m a bit confused here, the authors stated before that they used GOES-R to distinguish the volcanic clouds from the meteorological ones. But I see now that they also used GOES to detect the VTH? How and why? In the methodology section they were talking about SEVIRI and temperature profiles but now GOES appears.
Figures 2 and 4. The information contained in bias and MAPE are similar, I would report both in the same histogram maybe with 2 y-axes.
Section 4.2
Are you sure that the low clouds (below 5 km) are ash clouds? It could be useful to see the locations of these clouds.
Section 5.
Table 3. Please report also the absolute values of R, MAPE, STD and BIAS before and after including temperature profiles.
Section 6.
The whole section must be improved, it is messy and not informative.
I do not think that what they do here is to assess the” Sensitivity of SDA feature selection” as it is reported In the title, I would re-title it.
Lines 326-327: what does it mean? “… to assess the sensitivity of the SDA to the VTH.” I do not understand what they do here.
Figure 6 and 7. It is not possible to appreciate the features from the greyscale images. The features should be highlighted with some color or polygon, otherwise the figures and the paragraph are not useful at all.
Figure 6. What R4-11 are?
Reviewer 2 Report
Review of "Retrieving volcanic ash top height through combined polar orbit active and geostationary passive remote sensing" by Weiren Zhu et al.
General comments
In this new study, Zhu et al. discuss the estimation of cloud top heights of ash plumes of volcanic eruptions based on brightness temperature data from geostationary infrared imagers. The authors propose to use a hybrid deep learning algorithm based on the stacked denoise autoencoder (SDA), least squares support vector regression (LSSVR), and genetic algorithm (GA). The algorithm was trained and tested using SEVIRI observations of two volcanic eruptions, Eyjafjallajökull, Iceland, in April and May 2010 and Puyehue-Cordon Caulle volcanic complex, Chile, in June 2011. Ground truth data for the training of the algorithm was obtained from CALIOP lidar satellite observations.
The study was carefully conducted and shows very promising results, as the hybrid deep learning algorithm shows better performance in estimating the volcanic ash cloud top height than traditional methods (1D-VAR retrievals). The paper is well written, concise, and interesting to read. I would recommend it for publication in remote sensing subject to a few specific comments and corrections.
Specific comments
l27-28: Maybe clarify that "complex meteorological conditions" refers to clouds?
l35: Add a few references for the relevance of volcanic ash observations for climate effects and aviation safety?
l131-136: After separating the testing and the training data, did you consider cross-validation for training the hybrid deep learning model to avoid overfitting? Or has the model been trained only once using the full training data set?
l131-136: I got a bit confused whether you trained one "global" model using both the Eyjafjallajökull and the Puyehue data at the same time or did you train two separate models for both eruptions?
l168: Suggest to skip "third-generation" when referring to ERA-Interim, as the successor of ERA-Interim is ERA5, the fifth-generation reanalysis. Not sure what the fourth-generation would have been?
l171: ERA-Interim has an effective horizontal resolution of 80 km and not 31 km. Furthermore, ERA-Interim extends only up to 0.1 hPa and not up to 0.01 hPa. This sounds like descriptions of ERA-Interim and ERA5 have been mixed up here.
l182-183: Did you implement the algorithms used in this study yourself or did you use an existing software package for this? Which package did you use?
l186-190: I was wondering whether it would make sense to consider brightness temperature differences (BTDs) rather than brightness temperatures (BTs) as input for hybrid model? BTDs are often considered for the analysis of infrared observations, because variable background conditions are effectively removed by subtracting BTs of a background channel from a signal channel?
l191-202: It might be good to list the values of the hyperparameters found in this study so that the results may be reproduced by others.
l209: Equation (1) for the bias is missing a factor 1/n, I guess? Why did you calculate the bias from the absolute deviations? This way, we do not know whether the method is under- or overestimating the cloud top height.
l225-226: Add units for the bar charts. Perhaps add again the number of data points considered here?
l227-250: I was wondering whether the information presented here is redundant? It seems Figure 2 is already showing how the different components of the hybrid algorithm contribute to its performance?
l263-264: Did you actually check and compare the meteorological conditions (cloud situation) during the Eyjafjallajökull and Puyehue eruptions?
l263-274: Can you please clarify whether two separate deep learning models have been trained for the Eyjafjallajökull and the Puyehue eruptions or if the same model was used for both cases?
l309-312: Have multiple ash cloud layers actually been observed for the Puyehue eruption?
l314-324: Can you explain why there was quite significant improvement for the case of Eyjafjallajökull, but the improvement was much lower for the Puyehue? Is this also related to the meteorological conditions?
l340-343: I would suggest to add some arrows to Figs. 6 and 7 to point out the location of the volcanic ash cloud.
l350-351: Another option to select the hyperparameters would be "random search".
l402: Are you thinking of differences of day- and nighttime observations when you are referring to "different eruption times"?
l402-403: In the beginning of the manuscript, it was pointed out that the infrared observations are not very sensitive to the satellite observation angles?
l414-417: It might be good to see how the hybrid deep learning model performs on a test case for which it has not been trained at all.
l477-478: SVMs found many applications in remote sensing, perhaps add some references for these applications rather than for "nuclear technology"?
Technical corrections
l105: in _the_ appendix.
l176: fix "National Center for Atmospheric Research (ECMWF)"
l252: _traditional_
255: _true_ data (?)
l466: fix "where nk data" (?)
l495: introduce RBF acronym
Round 2
Reviewer 1 Report
The paper has been improved from the previous version, however there are several uncertainties which must be taken in account for such kind of study. This is the most critical part of the paper, once it is clarified the paper can be published. You can use machine learning in different ways, but the results depend on the input and output you provide to the algorithm: if the input parameters are uncertain, the outputs are uncertain too.
Major issues
- Please add a section with the uncertainties because there are several issues with collocations and cloud discrimination. Uncertainties given by the algorithm to estimate the VTH, uncertainty given by the space/time collocation of CALIOP/SEVIRI/ERA-Interim, uncertainty on the CALIOP algorithm, uncertainty given by the cloud discrimination and so on …
- The authors must explain what is the CALIO total attenuated backscatter that they included in the new version of the paper.
Other minor comments
Introduction: In my previous review I also suggested to include the radio occultation technique which is, at the moment, that one providing the highest accuracy in VTH estimation, but the author did not add it. I suggest to include it:
Cigala, V., Biondi, R., Prata, A. J., Steiner, A. K., Kirchengast, G., & Brenot, H. (2019). GNSS Radio Occultation Advances the Monitoring of Volcanic Clouds: The Case of the 2008 Kasatochi Eruption. Remote Sensing, 11(19), 2199.
Figure 1. I do not think that Point 2 is at 42S please correct it. Is it 42N?
Figure 2. The authors states that they use the CALIOP L2 cloud layer 5km but they show in figure 2 and figure 8 the total attenuated backscatter. They should also explain what is the backscatter product they use for these figures because the backscatter is not the L2.
Caption of Figure 8 refers to “… red circles …” which I can’t see
Figure 9 SEVERI is SEVIRI. Please highlight exactly the same area (or the cloud feature) in all the panels of Figure 9 and Figure 10 so we can compare and appreciate the differences.
Please double check the English.
